# The Effect of Solute Elements Co-Segregation on Grain Boundary Energy and the Mechanical Properties of Aluminum by First-Principles Calculation

**DOI:** 10.3390/nano14221803

**Published:** 2024-11-11

**Authors:** Xuan Zhang, Yuxuan Wan, Cuifan Chen, Liang Zhang

**Affiliations:** 1International Joint Laboratory for Light Alloys (MOE), College of Materials Science and Engineering, Chongqing University, Chongqing 400044, China; 2State Key Laboratory of Mechanical Transmission for Advanced Equipment, Chongqing University, Chongqing 400044, China

**Keywords:** grain boundary, solute atoms, co-segregation, GB energy, GB strength

## Abstract

The segregation of solute atoms at grain boundary (GB) has an important effect on the GB characteristics and the properties of materials. The study of multielement co-segregation in GBs is still in progress and deserves further research at the atomic scale. In this work, first-principles calculations were carried out to investigate the effect of Mg and Cu co-segregation on the energetic and mechanical properties of the Al Σ5(210) GB. The segregation tendency of Mg at the GB in the presence of Cu is characterized, indicating a preference for substitutional segregation far away from Cu atoms. Cu segregation can facilitate the segregation of Mg due to their mutual attractive energy. The GB energy results show that Mg and Cu co-segregation significantly decreases GB energy and thus enhances the stability of the Al Σ5(210) GB. First-principles tensile test calculations indicate that Cu effectively counteracts the weakening effect of Mg segregation in the GB, particularly with the high concentration of Cu segregation. The phenomenon of Cu compensating the strength of the GB is attributed to an increase of charge density and the formation of newly formed Cu-Al bonds. Conversely, Mg segregation weakens the strengthening effect of Cu on the GB, but it can increase the strength of the GB when high concentrations of Cu atoms are present in the GB. The ICOHP and Bader charge analysis exhibits that the strengthening effect of Mg is attributed to charge transfer with surrounding Al and Cu, which enhances the Cu-Al and Al-Al bonds. The results provide a further understanding of the interplay between co-segregated elements and its influence on the energetic and mechanical properties of grain boundary.

## 1. Introduction

Nanocrystalline alloys have attracted much attention due to their appealing mechanical properties, such as high strength, ductility, and resistance to tribological [1]. High-density grain boundaries (GBs) inhibit the dislocation movement and absorb the structure defects, leading to the high mechanical performance of these alloys [2,3]. Taking Cu as an example, the material with a grain size of 10 nm exhibits a strength of 1 GPa and a hardness that can reach 3 GPa, surpassing the 50 MPa strength observed in its coarse-grained counterpart [4]. However, the numerous GBs increase the energy and make the nanocrystalline alloys microstructurally unstable [5]. Substantial grain coarsening of nanocrystalline Cu can even happen at room temperature [6]. Also, as the grain size is reduced to a specific level, the GB migration of the nanocrystals will lead to a decrease in strength [7]. Solute GB segregation is a potent mechanism for increasing the stability and mechanical properties of nanocrystalline alloys by influencing the structure and chemical bond of the GB [8,9,10]. For example, nanocrystalline Ni with W segregation at the GBs shows a high stability at 600 °C compared to the pure nanocrystal Ni, which is stable only up to 300 °C under the same conditions [11,12]. In the work of Li et al. [13], nanostructure Fe was manufactured through severe plastic deformation, achieving a strength of up to 7 GPa and showing strong GB stability at 250 °C. The C segregation in the GBs is the main reason for its high strength and stable nanostructure, which substantially decreases the GB energy and inhibits the grain coarsening.

Numerous studies have demonstrated that solute GB segregation can increase the strength and stability of nanocrystalline Al alloys [14]. For instance, Valiev et al. [15] fabricated nanostructure Al alloys through a high-pressure torsion process, achieving a remarkable strength of 1 GPa, attributed to Mg solute segregation induced by deformation. Xu et al. [16] manufactured Al-Cu alloy with gradient nanostructures using surface mechanical grinding treatment at liquid nitrogen temperature, achieving a microhardness of up to 2.57 GPa. High-content Cu segregation in the GB is the main reason. Similarly, Jia et al. [17] manufactured a bimodal grain structured Al-Cu alloy prepared by an equal-channel angular pressing process. They found that the segregation of Cu induced by artificial aging increases the tensile strength of the alloy. In the work of Zha et al. [18], an Al-Mg alloy with a multimodal structure, manufactured by the equal-channel angular pressing method, shows enhanced thermal stability during recrystallization at 578 K. They found that the Mg solute segregation can suppress grain growth via diminishing GB energy and dragging GBs efficiently.

In Al-Cu-Mg alloys, Mg and Cu are the main solute elements and play crucial roles by exerting a strong solution-strengthening effect [19,20]. Their composite segregation (co-segregation) facilitates second-phase precipitation, which is another main strengthening resource [21,22,23,24]. Also, both experimental and simulation studies have consistently demonstrated that Mg and Cu atoms can form Al GB segregation simultaneously [25,26]. For example, Sha et al. [23] observed that there are 2.5 at% Mg and Cu atoms in Al-Zn-Mg-Cu alloys with ultra-fined grain. Additionally, their co-segregation can lead to the precipitation of the second phase in Al alloys [27]. However, using experimental methods, such as 3D atom probe technology, it is challenging to obtain detailed information on solute GB segregation and analyze their mechanism in influencing GB strength and stability at the atomic scale. With the development of simulation methods, first-principles calculations offer a more economical and efficient means of elucidating the solute GB segregation phenomenon at the atomic scale and understanding their effect on the GB energetic properties. Numerous studies have investigated the segregation effects of various solutes on the strength and stability of Al GBs using the first-principles calculation method [28,29,30,31,32]. For instance, Zhang et al. [33] explored the segregation behaviors of individual Mg and Cu atoms across various Al GBs. They found that both elements can simultaneously segregate at Al GBs, but the elements have different segregation propensity. Also, Mg decreases the strength while Cu strengthens the strength. Razumovskiy et al. [34] investigated the effect of alloying elements, including P, Ni, Mg, Zr, and B, on Al Σ5(210) GB cohesion energy. They found that Ni is the best GB enhancer, P plays the worst embrittlement role to the GB, and Mg has a neutral effect. Liu et al. [35] focused on Mg behavior at the Σ11(113) GB, showing that Mg atoms preferentially segregate to looser sites but play a weakening role. Recently, Yu et al. [36] examined the effect of Zn on different Al GBs. They determined the most energetically favorable site of Zn in GBs, calculated the stress strength of Zn-doped GBs, and analyzed the electronic structures and bonding characters, finding that Zn embrittles the studied GBs except Σ3(111) and Σ7(415) GBs. Despite these advances, most studies have focused on the segregation effects of individual elements on Al GBs.

Additionally, some investigations have concentrated on the solute co-segregation in the GB of metal alloys by simulation method [37]. For instance, in the work of Huang et al. [38], the co-segregation effect of metallic solutes (Al, Zn, Zr, etc.) and nonmetallic atoms (H, B, C, etc.) in the Cu Σ5(310) GB were investigated. Their findings revealed that the strengthening impacts of dopants and impurities were primarily governed by the electronic interactions existing between the host Cu atoms and the two dopant elements. Moreover, there is some relevant research on the co-segregation of solute in the Al alloys. For example, Xiao et al. [39] studied the co-segregation of Zr and Sc in the Al Σ5(210) GB and found that the co-segregation of these two solutes strengthens the GB. This effect is due to the movement of the Sc atom towards the GB, which decreases the distance between two grains and increases the charge accumulation around the GB plane, which results in stronger Sc-Al and Zr–Al bonds. In the study by Long et al. [40], the behavior of transition metal impurity atoms co-segregating with H atoms at three distinct Al GBs (Σ5(210), Σ11(113), and Σ3(111)) was examined using first-principles calculation method. The findings revealed that co-segregation energy is significantly influenced by the distinctive segregation patterns of the impurity atoms and the nature of their interactions with H. While molecular dynamics studies have explored Mg and Cu co-segregation [41], they did not give the mechanism of how the two elements’ co-segregation influence the energetic and mechanical characteristics of the Al GB on the atomic scale and the atomic interaction between the Mg and Cu atoms. Therefore, it is meaningful to elucidate how the co-segregation of Mg and Cu impacts the strength and stability of the Al GB.

Based on the above introduction, the influences of the co-segregation of Mg and Cu on the Al Σ5(210) GB were examined using the first-principles calculation method. In the first part, the segregation energy was calculated to decide the segregation positions of Mg in the Al GB with Cu segregation. Subsequently, the segregation energy and binding energy of different Mg Cu co-segregation combinations were calculated, and their effect on the GB stability was assessed by calculating the GB energy. In the second part, the theoretical strength and fracture energy of the GB were calculated to analyze the co-segregation effect on the GB. In the third part, the differential electron charge density, density of state, and Crystal Orbital Hamiltonian Populations of the main atomic bonding in the fracture plane of GBs were computed and used to elucidate the underlying mechanisms of the co-segregation of Mg and Cu on the mechanical performance of the GB.

## 2. Materials and Methods

### 2.1. First-Principles Calculation

The first-principles calculations for this study were all performed utilizing the Vienna ab initio Simulation Package (VASP), employing the projector-augmented wave (PAW) approach [42,43]. For optimal accuracy and efficiency, the cutoff energy was set as 350 eV for all simulations. The conjugate gradient method was utilized to perform the relaxation work. The force convergence criterion was set under 0.01 eV/Å. A convergence threshold of 10^−6^ eV was applied to the model energy. A *k*-point mesh of 6×5×3 within the Monkhorst–Pack scheme was sufficient for calculations. In addition, the 3×3×3 mesh was employed in the Al bulk calculation using a 108-atom supercell with the convergence criteria defined above. All atoms and the supercell vectors were fully relaxed. The calculated equilibrium lattice constant at 0 K is 4.049 Å, closely matching the experimental result of 4.036 Å at 20 K [44]. Then, the charge density and finial energy of models were calculated by the self-consistent static electronic relaxation method. In addition, the differential electron charge density (DECD) of GB supercell is commonly used to explore point-defect interactions between matrix and solute atoms. The DECD for the relaxed model is obtained by subtracting the charge density obtained by just one step electron calculation from the charge density calculated above, which can be represented by Equation (1) [45]:(1)$$\rho_{DECD}\left({Al}_{x-y}{Mg}_{y}{Cu}_{z}\right)=\rho \left({Al}_{x-y}{Mg}_{y}{Cu}_{z}\right)-\rho \left({Al}_{x-y}\right)-\rho \left({Mg}_{y}\right)-\rho \left({Cu}_{z}\right)$$
where $\rho \left({Al}_{x-y}{Mg}_{y}{Cu}_{z}\right)$ represents the charge density of the supercell containing Mg and Cu after full electronic relaxation, while $\rho \left({Al}_{x-y}\right)$, $\rho \left({Mg}_{y}\right)$, and $\rho \left({Cu}_{z}\right)$ refer to the charge density of the same supercell containing only Al, Mg, or Cu, respectively, from a single step of electronic relaxation. In the supercell, x−y corresponds to the number of Al atoms, and y is the number of Mg atoms.

### 2.2. Grain Boundary Model

The Al Σ5[001](210) GB, which was observed in the experiment [46], was chosen as the supercell model and constructed by the coincidence symmetry lattice method [2,47,48,49]. The pristine GB model is a supercell that contains 40 layers, and each layer is composed of 2 atoms. Half of the layers rotate around the [001] axis with an angle of 53°; then, the GB are formed. As shown in Figure 1a, two GBs were positioned at the center and bottom of the supercell to maintain periodicity. The supercell dimensions were given as (2,√2,√5)a, where a was the lattice constant of Al fcc supercell. The separation distance is sufficient to prevent atomic interactions between them. For the Mg and Cu co-segregation model, according to related experimental [50] and calculation investigations [51], the Mg atoms are prone to form substitutional segregation, while Cu atoms prefer to form interstitial segregation. Additionally, the previous work [33] demonstrates that Cu atoms exhibit a higher segregation tendency than Mg atoms, indicating a stronger segregation tendency for Cu in GB. Hence, as shown in Figure 1b, Cu atoms are firstly segregated at the interstitial site. The structural unit of the GB is disrupted by the segregation of Cu atoms at the interstitial site, creating 6 possible sites for Mg segregation. Then, as shown in Figure 1c, the Mg atom is set in site 6 to illustrate the solutes’ co-segregation model. All the GB supercells in this work were performed with relaxation calculation to obtain the final stable structure and static calculation to acquire the final energy. The charge density and atomic structure diagrams of the GB in the present work were generated using the VESTA package [52].

### 2.3. Mechanical Property Calculation

First-principles tensile test calculation was implemented on the GB models with and without segregation to measure their GB mechanical properties. Generally, there are three main methods to implement the FPTCC. In the first method, the Poisson effect is considered, and the applied strain is evenly distributed across the entire GB supercell. During each stretching step, the lattice is evenly elongated in the direction of strain application. In the second scheme, the fracture planes in the GB are set in advance, and the distance between them increases during the stretching process. Afterward, full atomic relaxation was performed on the GB model. Rigid grain shift (RGS) is another method, excluding atomic relaxation, and is similar to the second method, except for the absence of relaxation. All of these methods can be used to measure the strength of the GB, and they yield similar relative results for tensile strength and fracture energy. However, the first method does not set the pre-crack and fracture plane in advance, meaning it cannot ensure that the fracture occurs in the GB plane. Additionally, considering the Poisson effect and atomic relaxation requires substantial time and computational resources. Moreover, when performing the relaxation process on fractured GB, achieving convergence is challenging, which may even lead to calculation failure [53,54]. Moreover, according to the work of Tian et al. [55], whether the Poisson effect is taken into account has a negligible effect on the fracture energy and the strength of the GB. As a result, the strength of the GB in this study was determined using the RGS technique during the analysis of the tensile test.

## 3. Results

### 3.1. Segregation Site of Mg

As mentioned above, Cu has a stronger segregation tendency to the Al Σ5(210) GB than the Mg atom, indicating that Cu forms interstitial segregation first in the GB. Then, the Mg atoms substitute Al atoms and form segregation in the GB. In addition, the Cu atoms disrupt the unsegregated GB unit, thereby influencing the Mg segregation site. Hence, to analyze the co-segregation of Mg and Cu, identifying the energetically favorable site of Mg in the GB with existing Cu segregation is indispensable. The segregation energy of Mg at various positions was computed by Equation (2) to determine the most energetically favorable site [45]:(2)$$E_{seg}=\left(E_{GB}^{Mg+Cu}-E_{GB}^{Cu}\right)-\left(E_{bulk}^{Mg+Cu}-E_{bulk}^{Cu}\right)$$
where $E_{GB}^{Mg+Cu}$ and $E_{GB}^{Cu}$ represent the total energies of supercells containing Mg and Cu segregation and containing Cu segregation, respectively. $E_{bulk}^{Mg+Cu}$ refers to the total energy of a 108-atom Al fcc model with dissolved Cu and Mg atoms. $E_{bulk}^{Cu}$ signifies the energy of the same models with Cu solutes. The number of the segregation solutes varies from one to four, for there are four sites in the GB to form segregation. The negative value indicates the solute segregation at the GB is energy favorable, while a positive value indicates it is not.

Figure 2a illustrates the $E_{seg}$ of Mg in different sites of the Al GB, it can be observed that the values of $E_{seg}$ at site 1, 2, and 5 are positive and almost equal, about 0.23 eV. The values at sites 3, 5, and 6 are −0.53, −0.42, and 0.85 eV, respectively, indicating that Mg is more prone to segregate at sites 3 and 6 of the one Cu-segregated GB. The segregation behavior of the Mg atom is identical at site 6 and site 3 of the pristine Al GBs. However, the Cu segregation changes the structure characteristic of the Al GB, leading to the discrepancy of segregation tendency. Hence, when Mg forms segregation at the GB with Cu, site 6 is preferred over site 3. The Mg atom at site 6, located in different planes around the [001] axis from the Cu atom exhibits the lower segregation energy. Therefore, on the view of segregation energy, Mg atoms are prone to form segregation at the sites that is far from the interstitial Cu segregation. With four interstitial and substitutional sites available in the Al GB interface, Mg and Cu atoms can occupy zero, one, two, three, or four of these sites, leading to a total of 25 possible Mg and Cu segregation combinations. GB models with Cu segregation alone are constructed and investigated to compare the energetic and mechanical characteristics of the GB with different segregations. According to compare these situations, the Mg segregation order is decided. The Cu and Mg co-segregated GB models are constructed and shown in Figure 3. Figure 3a demonstrates the different Cu-segregated GB models. To show the Mg segregation order in Cu-segregated GB already, the two Cu-segregated GB are chosen as an example and illustrated in Figure 3b.

### 3.2. Energy Properties of GB

To evaluate how solute segregation coverage affects the energetic characteristics of the Al Σ5(210) GB, the supercells containing various combinations of Mg and Cu atom concentrations were built. Additionally, GB energy is a crucial indicator that reflects the stability of the GB. Therefore, to assess the effect of how the co-segregation of Cu and Mg affects the stability of the GB, the GB energy values at varying solute concentrations were calculated using Equation (3) [33]:(3)$$\gamma_{GB}=\frac{E_{GB}^{Mg+Cu}-N_{Al}E_{Al}-N_{Mg}E_{Mg}-N_{Cu}E_{Cu}}{2S}$$
where the $E_{GB}^{Mg+Cu}$ denotes the energy of the GB supercells containing both Mg and Cu segregation, and $E_{Mg}$, $E_{Cu}$ is the energy of a single Mg, Cu atom. $N_{Al}$, $N_{Mg}$, ${and\;N}_{Cu}$ denote the amounts of corresponding atoms in the GB model. S represents the GB area, and the subscript “2” indicates that there are two GBs.

To obtain the segregation propensity of the Mg atom in the Cu-segregated Al GB, the $E_{seg}^{Mg}$ were calculated by Equation (2). Also, the interaction between Mg and Cu was measured by calculating the binding energy. When the binding energy is positive, it means the two atoms attract each other, while a negative binding energy indicates that they repel each other. The value can be obtained by Equation (4) [40]:(4)$$E_{GB}^{Mg+Cu}=E_{GB}^{Mg}+E_{GB}^{Cu}-E_{GB}^{Cu,Mg}-E_{GB}$$

First, the pristine GB energy (value of $\gamma_{GB}$) is calculated, and the value is 0.518 J/m^2^, aligning well with other simulation results of 0.518 J/m^2^ [51], 0.497 J/m^2^ [28], and the experimental result, 600 J/m^2^ [56]. The $\gamma_{GB}$ results of the GB with the co-segregation of various Mg and Cu are demonstrated in Figure 2b. The addition of Mg atoms to the Cu-segregated GB is found to further reduce the value of $\gamma_{GB}$. For example, the value of $\gamma_{GB}$ with one Cu segregation is 0.381 J/m^2^, but it decreases to 0.3 J/m^2^ with one Mg + one Cu co-segregation. On the other hand, based on the GB with one Mg segregation, the value of $\gamma_{GB}$ decreases as the increment of Cu atoms, from 0.473 to −0.238 J/m^2^. The four Mg addition at the GB with four Cu segregation evenly makes the GB energy down to −0.478 J/m^2^, it is almost twice a decrease than the pristine GB energy, 0.518 J/m^2^. Therefore, the co-segregation of Mg and Cu atoms at GB has a better effect on increasing the stability of the GB, which can be ascribed to the further decrease in free volume and the strain energy release of the GB brought by Mg and Cu co-segregation [57,58].

Figure 2c plots the segregation energy ($E_{seg}^{Mg}$) of the Mg atom in the one Cu- to four Cu-segregated GBs. First, $E_{seg}^{Mg}$ is negative in all cases, with the largest value reaching −3.38 eV when four Mg and one Cu form co-segregation at the Al GB. This indicates that Mg and Cu can form segregation at all combinations of concentration. Furthermore, it can be observed that the segregation tendency of Mg decreases with the increase in Cu segregation. For instance, the $E_{seg}^{Mg}$ in the GB with one Cu segregation decreases from −6.67 to −3.38 eV when the number of Mg atoms increases from one to four, indicating that the Mg segregation tendency decreases as Mg segregation increases. Additionally, increasing Cu segregation at the GB enhances the segregation propensity of Mg atoms, meaning that Cu promotes the segregation of Mg. For example, the value of $E_{seg}^{Mg}$ of two Mg increases from −4.48 to −11.37 eV as Cu concentration increases from one to four. Finally, the promoting effect of Cu on Mg segregation increases with the number of Cu atoms, as observed from the decreasing $E_{seg}^{Mg}$ values.

Figure 2d illustrates the binding energy of Mg and Cu in the GB with Cu segregation. The binding energy value reflects the repulsion and attractive interaction between the Mg and Cu. The positive values indicate that the subsequent Mg segregation has an attractive interaction, meaning that the Cu and Cu co-segregation is energy favorable. The binding energy is a zero value for Mg, which means there is only Cu segregation at the GB. The binding energy for the 1st Cu is not applicable, as there is only one Cu atom. Additionally, the binding energy shows an increasing trend from 0 to the 2nd Mg, which can be attributed to the short distance between the two solutes. The main reason for the decrease in binding energy is the large distance between the 3rd Mg atom and the 1st Cu atom. It can also be observed that the binding energy of all Mg and Cu combinations is positive, indicating that they attract each other. Therefore, the segregation of Mg can be promoted by Cu atoms in the Al GB, which explains the energetically favorable Mg and Cu co-segregation.

### 3.3. Mechanical Properties of GB

To explore the effects of different co-segregation combinations of Mg and Cu on the mechanical characteristics of the Al Σ5(210) GB, the theoretical strength and fracture energy were calculated using the first-principles tensile test calculation. These calculations were carried out with the fully relaxed GB models with different Mg and Cu co-segregation. The supercells were rigidly separated from 0 to 6 Å during the tensile test. Subsequently, the energies of the supercells were fitted using the equations below.

The tensile tests were employed by the rigid grain shift method to ensure accuracy and save calculated resources. Equation (5) [59] was used to compute the separation energy of the GB:(5)$$E_{sep}=\frac{E_{GB}^{x}-E_{GB}}{2S}$$
where $E_{sep}$ is the separation energy. $E_{GB}^{x}$ denotes the energy across variously segregated GB models, corresponding to a separation distance of x. $E_{GB}$ indicates the energy of the same model when there is no separation. The relationship between the $E_{sep}$ and x can be analyzed using Equation (6), as introduced by Rose et al. [60]:(6)$$f\left(x\right)=E_{frac}-E_{frac}\left(1+\frac{x}{\lambda }\right)e^{\left(-\frac{x}{\lambda }\right)}$$
(7)$$E_{frac}=\frac{E_{\infty }-E_{GB}}{2S}$$
where the $E_{frac}$ is the fracture energy that can be obtained by Equation (7) when the two fractured planes are far enough. λ denotes the critical separation distance [61]. The derivative of $f\left(x\right)$ is used to calculate the theoretical strength, as shown in the Equation (8):(8)$${f\left(x\right)}^{\prime }=\frac{E_{frac}x}{\lambda^{2}}e^{-\frac{x}{\lambda }}$$

Also, when the separation distance is λ, the theoretical strength is the maximum that can be calculated by Equation (9):(9)$${ð}_{max}={f\left(x\right)}^{\prime }=\frac{E_{frac}}{\lambda e}$$

The mechanical response of the GB with one Cu + x Mg and two Cu + x Mg co-segregation is illustrated in Figure 4. The results of only Cu- or Mg-segregated GBs, the pristine GB, and the Al bulk are also exhibited as a comparison. Figure 4a,b show perfect Rose fit curve diagrams between separation energy and the separation distances of the different co-segregated GBs. For all the GB systems, the separation energy gradually rises as the separation distance grows, then upwards to the largest value, and it eventually maintains it, which is the final separation energy. The final value, called the fracture energy, has been collected and summarized in Table 1. For pristine GB and bulk, the fracture energy is 1.75 and 2.15 J/m^2^, respectively, matching the experimental results of 1.83 and 2.16 J/m^2^ [54]. Additionally, a negative relationship between fracture energy and Mg concentration can be observed in the case of a GB with one Cu and two Cu segregations. For instance, the fracture energy value of the one Cu + one Mg-segregated GB is 1.73 J/m^2^, exceeding the 1.67 J/m^2^ fracture energy of the GB with one Cu + three Mg segregation. The fracture energy also exhibits a maximum value of 1.99 J/m^2^ when one Mg + two Cu segregates to the Al GB, which is higher than that of other conditions with the two Cu segregation.

The strength performance distribution of one Cu + x Mg and two Cu + x Mg co-segregated Al GB as the increasing separation distance are displayed in Figure 4c,d. For the GB with different segregations, the tensile strength sharply increases with increasing separation distance, which reaches a maximum value at the critical separation distance and then decreases to zero. To analyze the influence of the co-segregation of Mg and Cu on the mechanical characteristics of the Al GB, the peak values of the Al GB with various combinations of two elements are collected in Table 1. The peak strength value of the one Cu-segregated Al GB is 11.25 GPa, which exceeds the strength of the unsegregated Al GB and agrees well with the experimental work of 10.86 GPa [51]. The strength of the pristine GB is 10.56 GPa, which is lower than that of Al bulk (12.05 GPa). This indicates that the GB causes the Al to become weaker. Additionally, the strength of the GB with one Mg is 10.26 GPa, which falls below the strength of the pristine Al GB. These findings imply that Cu segregation strengthens the Al GB, whereas Mg has the opposite effect, and the effect increases with their increased segregation concentration. This finding aligns with that of other researchers [33,62], who found a similar phenomenon of Mg/Cu segregation in the multiple Al GBs, such as the Σ5(210) GB, Σ5(310) GB, Σ9(221) GB, and Σ11(332) GB. These results are consistent with the finding in the experimental results, which found that the Mg segregation induced an intergranular fracture phenomenon in Al-Mg alloys [63]. The theoretical strength of the one Cu-segregated GB decreases as the increasing Mg segregation. Also, the strength of the one Cu-segregated GB decrease to 10.56 GPa, which is equal to that of the pristine GB, is caused by the segregation of two Mg atoms. The strength of the one Cu + three Mg-segregated GB further reduces to 10.27 GPa, indicating that the weakening effect of three Mg segregation outweighs the strengthening effect of one Cu. The strength of one Cu + four Mg-segregated GB decreases by 1.44 GPa, about a 12.7% reduction compared to the GB with one Cu segregation. Similarly, the GB strength drops with the increasing number of Mg solutes in the two Cu-segregated GB. For instance, the lowest value of the GB with two Cu segregation caused by a four Mg addition is 11.18 GPa, about an 8.9% decrease in strength. The higher value compared to the pristine GB indicates that two Cu segregation in the GB can compensate for the strength reduction induced by four Mg segregation in the GB. According to our previous work [33] and other studies [54,62], it is easy to understand that Mg weakens the GB for causing the GB expansion and charge depletion, while Cu strengthens the GB for causing the charge accumulation. Therefore, it is understandable that Mg can counteract the strengthening effect of Cu on GB strength, and Cu segregation increases the Al GB fracture energy and strength. Overall, in the one Cu- and two Cu-segregated GB, the strength decreases with increasing Mg segregation.

In Figure 5, the distribution of separation energy and theoretical strength is presented for the GB featuring co-segregations of three Cu + x Mg and four Cu + x Mg at different separation distances. As demonstrated in Figure 5a,b, the different phenomena of separation energy are observed in the three Cu + x Mg- and four Cu + x Mg-segregated GB, where Mg does not reduce the fracture energy; rather, it can increase the fracture energy. For instance, the four Mg segregation increases the fracture energy of the four Cu-segregated GB from 2.15 to 2.29 J/m^2^. Furthermore, the strength of the GB with three Cu + x Mg and four Cu + x Mg co-segregation is, respectively, depicted in Figure 5c,d. Their peak values of the strength are plotted in Figure 5e to illustrate the variation in the strength of Cu-segregated GBs as the Mg segregation increases. It can be found that the strength weakening effect of Mg segregation in the GB with high Cu segregation is lower compared to that in the GB with low Cu segregation. For instance, four Mg addition in the GB with three Cu segregation only causes a 4.9% reduction in the strength, which is lower than the 8.9% decrease caused by four Mg in the GB with two Cu segregation. The peak strength value of the four Cu-segregated GB even demonstrates an increasing trend with the increase in Mg atoms, which indicates that Mg slightly increases the strength of the GB in this case. For example, the addition of four Mg atoms enhances the GB strength, elevating it from 13.34 to 13.50 GPa. The observed phenomenon where Mg increases both fracture energy and strength indicate the potential existence of an alternative mechanism in the present study.

In addition, the critical separation distance values of tensile strength are collected in Table 1. It is found that the critical separation distance (λ) increases with the increment of Mg segregation in the Al GB with x Cu segregation, suggesting that Mg increases the rigidity of the GB. Conversely, Cu exhibits an opposite trend, indicating that Cu decreases the rigidity of the GB. We attribute this variation in stiffness to changes in the bonding nature induced by Mg and Cu segregation. The largest λ value is observed in the GB with four Cu + four Mg co-segregation, indicating increased rigidity in this case. As analyzed above, this case demonstrates a strength-enhancing effect of Mg segregation in the Al GB with four Cu segregation, which is attributed to the combined impacts of Mg and Cu segregation.

## 4. Discussion

### 4.1. Charge Density Analysis

The charge density calculation is an effective method to qualitatively evaluate the effect of charge accumulation and charge depletion on the atomic bond strength in the GB plane. To understand how Cu compensates for the strength reduction effects of Mg in the Al Σ5(210) GB with two Cu + one Mg co-segregation as well as Mg’s strengthening effect in the four Cu + x Mg co-segregated Al GB, the charge density and DECD in the [001] direction for the GB with two Cu + one Mg co-segregation and one Mg segregation cases were calculated and analyzed. Furthermore, we analyzed the charge density for both the four Cu-segregated and the four Cu + four Mg co-segregated GBs across various separation distances, including their DECD as projected onto the [001] plane.

The charge density distribution and the structure on two GB planes of one Mg-segregated GB and two Cu + one Mg-segregated GBs are illustrated in Figure 6. As shown in Figure 6(a1–a3), it is obvious that there are strong charge accumulations in the Al (2)-Al (2′) and the Al (1)-Al (2) bonds in [001] plane, which indicates the presence of strong electronic interaction. These two bonds also mainly contribute to the GB’s strength. The plane [002] with Mg segregation that surrogates the Al (1) shows dissimilar conditions, with only obvious charge density existing in the Al (2) and Al (2′) bond. Additionally, there is no obvious charge accumulation in Mg-Al (2) for Mg less valence electron. In addition, as illustrated in Figure 6(b1–b3), Cu segregation eliminates the low charge density area. Furthermore, the repulsion caused by the Cu atom to the Al (1) atom leads to a decrease in bond length from 2.94 to 2.88 Å, resulting in stronger charge accumulation in the Al (1)-Al (2) bond, as shown in Figure 6(a4,b4). Despite the GB expansion due to Cu repulsion, the charge density of the Al (2)-Al (2) bond in the [001] plane remains unaffected, and Cu forms new bonds with neighboring Al atoms, such as Al (3), which have stronger charge accumulation than that of Cu-Al bonds in the one Cu-segregated GB. Overall, the addition of Cu can compensate for the weakening effect of Mg by forming new bonds.

Figure 7 demonstrates the charge density distribution at different separation distances, GB structure, and DECD of [001] projection of the unsegregated Al GB, four Cu-segregated Al GBs, and four Cu + four Mg-segregated Al GBs. Figure 7(a1–a3,b1–b3) show the cases of GB with four Cu segregation and pristine GB, which have been discussed in our previous work [33]. Compared to the pristine GB, there is strong charge accumulation in the newly formed Cu-Al (3)/Al (3′) bond, Al (2)-Al (2′) bond, and Al (1)-Al (2)/Al (2′) bond, which contribute mainly to the strength of GB. Then, as illustrated in Figure 7(b3), the charge density in these bonds in the GB plane drops as the separation distance increases. The low charge density region traverses the whole GB interface at the separation distance of 1.6 Å. Nonetheless, with the segregation of four Mg in site 1 of GB, it can be found the GB structure expansion is relieved slightly around the [210] direction. To illustrate, the bond length of Al (2)-Al (2′) shown in Figure 7(b1,c1) decreases from 2.84 to 2.80 Å. Moreover, the DECD of Al GB with four Cu and four Cu + four Mg segregation in [001] projection is calculated and illustrated in Figure 7(b2,c2). Stronger charge accumulation in the Al (2)-Al (2′) bond of the Al GB with Mg segregation than that of the same bond in the GB without Mg can be observed, meaning the higher bond strength. From this point, the Mg addition is positive to the strength in this case. However, compared to the Al (1)-Al (2′)/Al (2′) in Figure 7(c3), there are few charge accumulations observed between Mg and Al (2)/Al (2′) in Figure 7(b3), implying that the Mg-Al bonds in GB with four Cu + four Mg co-segregation are weaker compared to that of GB without Mg segregation. Therefore, it is difficult to explain the results that the addition of Mg can either strengthen or weaken the GB with four Cu segregation just based on GB structure and charge density.

### 4.2. Partial Density of States

To further investigate the Mg effect on bond characteristics of four Cu-segregated Al Σ5(210) GBs, the partial density of states (PDOS) of segregation atoms and neighboring Al atoms were computed. Figure 8 presents the PDOS of Mg, Cu, and the adjacent Al atoms in the Al GB with four Cu and four Cu + four Mg segregation. These calculations provide insights into the characteristics of chemical bonds. Figure 8a,b show the electron distributions in s and p orbits for the Al (2) atoms in the four Cu-segregated GB and four Cu + four Mg-segregated GB, respectively. Notably, the nearly identical electron distributions in the s and p orbitals of the two atoms in Figure 8a indicate a significant electronic interaction. A similar bonding condition is observed in Figure 8b but with a higher bond strength in the case of four Mg + four Cu co-segregation. For instance, for Al (2), the s and p electron densities exhibit elevated levels between the energy ranges of −10 to −5 eV and −2 to 4 eV, signifying a stronger Al (2)–Al (2) bond compared to that found in the Al GB with solely four Cu atoms segregated. Figure 8c illustrates the electron distributions within the s and p orbitals for the Al (1) and Al (2) atoms. A similar distribution of electrons in both orbitals can be observed, indicating the sturdy chemical bond exists between them. In addition, Mg atoms substitute Al (1) atoms in the GB and form a co-segregated GB. The reduced overall electron distribution for Mg, compared to Al (2), indicates charge transfer between Mg and the surrounding Cu and Al atoms, as shown in Figure 8d. Additionally, the number of hybridization peaks between Mg and Al (2) is reduced, and their intensity is diminished as a consequence of the lower number of valence electrons present in Mg. This suggests that the Al (1)–Al (2) bond is stronger than the Mg-Al (2) bond.

To qualitatively compare the strength of two Cu-Al bonds, Figure 8e,f display the PDOS for the electron distributions in s and p orbitals of Cu and Al (3) atoms in two segregated GB cases. As demonstrated in Figure 8e, observable formants in the s electrons of two atoms at various energies for the similar distribution of the PDOS, such as −6.85 eV and −9.70 eV, suggests high electronic interaction between Cu solute atoms and surrounding Al atoms. Figure 8f illustrates the PDOS for the electron in s and p orbits of Cu and Al atoms in the four Mg + four Cu co-segregated Al GB. The chemical bond situation is similar to the previous case. However, the p electron density of Cu in Figure 8f is higher than that in Figure 8e in the range from −2 to 2.5 eV. A similar increase is observed in the Al (3) atom in the four Cu + four Mg co-segregated Al GB. Consequently, except for the Al (1)-Al (2) bond being stronger compared to the Mg-Al (2) bond, the other bonds (e.g., the Al (2)-Al (2′) bond) in the Al GB with four Cu segregation are weaker than that of Al GB with four Cu + four Mg co-segregation.

### 4.3. Crystal Orbital Hamiltonian Populations and Bader Charge

The charge density and PDOS cannot provide quantitative results for bond strength in the GB plane. In addition, the charge density does not directly reflect bond strength and can only give bond information in the same plane, whereas many bonds consist of atoms from different planes. Therefore, to quantitatively characterize the relative strength contributions of the primary bonds along the fracture plane of four Cu-segregated Al GBs and four Cu + four Mg co-segregated Al GBs, Crystal Orbital Hamiltonian Populations (COHP) and integrated COHP (ICOHP) values were computed using the software Lobster 5.0.0 [64] and are presented in Figure 9 and Table 2, respectively.

Antibonding states, indicated by a negative COHP value below the Fermi level, signify overlapping bond orbitals of two atoms. This scenario results in a system with reduced stability and weaker bonds. ICOHP values serve as indicators of bond strength, with smaller values corresponding to stronger and more stable chemical bonds. These bonds include Al-Al bonds, Mg-Al bonds, and Cu-Al bonds at the four Cu-segregated GB along the fracture path as well as the corresponding bonds in the four Cu + four Mg co-segregated GB. As shown in Figure 9a, both the Al (2)-Al (2′) and Al (1)-Al (2) bonds exhibit antibonding regions beneath the Fermi level, suggesting electron instability. Conversely, in the GB with four Cu + four Mg co-segregation, the first bond shows no antibonding regions, as illustrated in Figure 9b, which implies that the stability of bonds is enhanced by the Mg atom. The ICOHP value of this bond is −2.42 eV/bond, which is lower than that of the corresponding bonds in Figure 9a, −1.65 eV/bond. This indicates an increased bond strength. Additionally, the Al (1)-Al (2) bond is stronger than the Mg-Al (2) bond in Figure 9b for its ICOHP value of −1.05 eV/bond, which is lower compared to the −0.14 eV/bond for the Mg-Al (2) bond. Notably, the Cu-Al (2) bond shows a higher value of −1.75 eV/bond compared to the same bond in Figure 9b, which is −2.08 eV/bond, suggesting greater strength in the latter bond. Furthermore, the cumulative ICOHP value of the main bonds in the four Cu-segregated GB is higher compared to that of the corresponding bonds in the GB with four Cu + four Mg segregation, which suggests the stronger cohesion strength of the four Cu + four Mg-segregated GB.

Additionally, Table 2 provides detailed information on the Bader charge of the primary atoms (e.g., Al (1), Al (2), Cu, Mg, etc.) that form the main strength-contributed bonds. Bader charge calculations were performed using the Bader Charge Analysis package [65] to obtain the reliable approximation charge transfer among the Mg/Cu and matrix Al atoms. This method enables the determination of changes in the valence electrons of Mg and Cu after segregation from the bulk to the GB. Cu atoms have higher electronegativity than Al atoms, leading to electron transfer from Al to Cu. The greater the Bader charge is, the greater the amount of charge transfer that has occurred. Conversely, the Mg atom, with lower electronegativity, transfers valence electrons to neighboring Al atoms, and an elevated Bader charge indicates a small charge transfer effect. A comparative analysis reveals that the valence electrons transfer among segregated solutes and Al atoms in the GB with four Cu + four Mg co-segregation are higher than that of the corresponding atoms in the GB with four Cu segregation. For example, the Bader charge of Al (2) and Al (3) in the GB with four Cu + four Mg co-segregation is 2.72 e and 2.83 e, respectively, which exceeds that of the same atoms in the GB with four Cu segregation. This pattern is consistent for Cu atoms in both cases. Notably, the Bader charge of Mg is 0.52 e, and the corresponding Al (1) atom is 2.56 e (valence electrons of Mg and Al being 2 e and 3 e, respectively). This indicates that the charge transfer between Mg and Al is stronger compared to that occurring between Al and Al. Therefore, the primary factor contributing to the increased bonding strength is the charge transfer effect of Mg with its surrounding atoms.

## 5. Conclusions

The first-principles calculation is a versatile method to investigate how the co-segregation of Mg and Cu affects the energetic and mechanical properties of the Al Σ5(210) GB. GB energy, GB segregation energy, and the mechanical strength of the GB were calculated. The strengthening mechanism of Mg and Cu co-segregation was analyzed by the calculation of the differential charge density, PDOS, COHP, and integrated COHP. The main findings are summarized as follows:
(1)By calculating the segregation energy of Mg at different positions of the Cu-segregated Σ5(210) GB, the segregation preference and the preferred location of Mg at the GB were identified. The results indicate that Mg tends to substitute the Al atom and form segregation at the position of the GB that is far from the interstitial segregation of Cu.(2)Mg and Cu co-segregation significantly alters the energy properties of the GB. GB energy calculations exhibit that Mg and Cu co-segregation notably decreases GB energy, which further improves the stability of the Σ5(210) GB compared with Mg or Cu segregation alone. The segregation of Cu enhances Mg segregation, and this effect strengthens with increased Cu concentration. According to the analysis of the binding energy between Mg and Cu, the primary factor driving Cu to facilitate Mg segregation is the energy of mutual attraction between them.(3)First-principles tensile test calculation performed on the mechanical properties of the GB indicate that Cu can inhibit the weakening effect caused by Mg segregation at the Al GB. With the increased number of segregated Cu atoms, the inhibition becomes more obvious. Mg weakens the strengthening effect of low concentration Cu, and the GB strength decreases gradually with the increase in Mg atoms. However, Mg strengthens the GB in the case of a GB with a high concentration of Cu atoms.(4)The results of the charge density, PDOS, and COHP calculations indicate that the compensating effect of Cu on the GB strength is mainly ascribed to the increment of charge density at the GB core region, the newly formed Cu-Al bonds, and stronger Al-Al bonds. The strengthening effect of Mg on the GB with a high concentration of Cu can be attributed to the fact that the segregation of Mg slows the expansion of the GB, although Mg reduces the charge density of the GB. Also, the segregated Mg atoms increased the Cu-Al and Al-Al bonding involving charge transfer with surrounding atoms.


## Figures and Tables

**Figure 1 nanomaterials-14-01803-f001:**
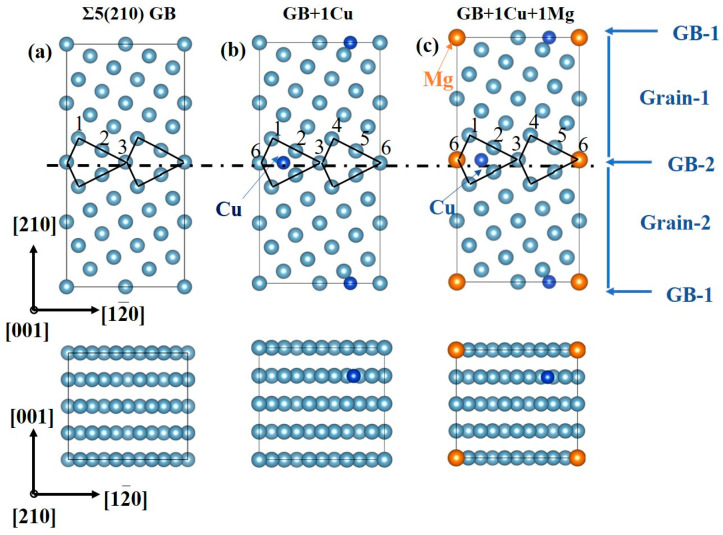
The atomic structure illustration of the (**a**) unsegregated Al Σ5(210) GB, (**b**) 1 Cu-segregated Σ5(210) GB, and (**c**) 1 Cu + 1 Mg co-segregated Σ5(210) GB.

**Figure 2 nanomaterials-14-01803-f002:**
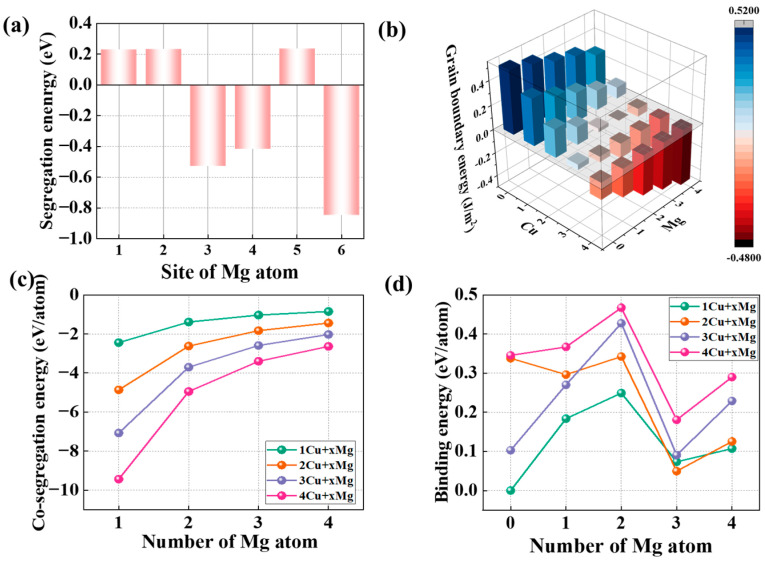
(**a**) The segregation energy of Mg in different sites of the Al Σ5(210) GB, (**b**) the GB energy with different Mg and Cu co-segregation, (**c**) the segregation energy of Mg in the Al GB with different Cu segregation already, and (**d**) the binding energy of Mg and Cu in the Al GB.

**Figure 3 nanomaterials-14-01803-f003:**
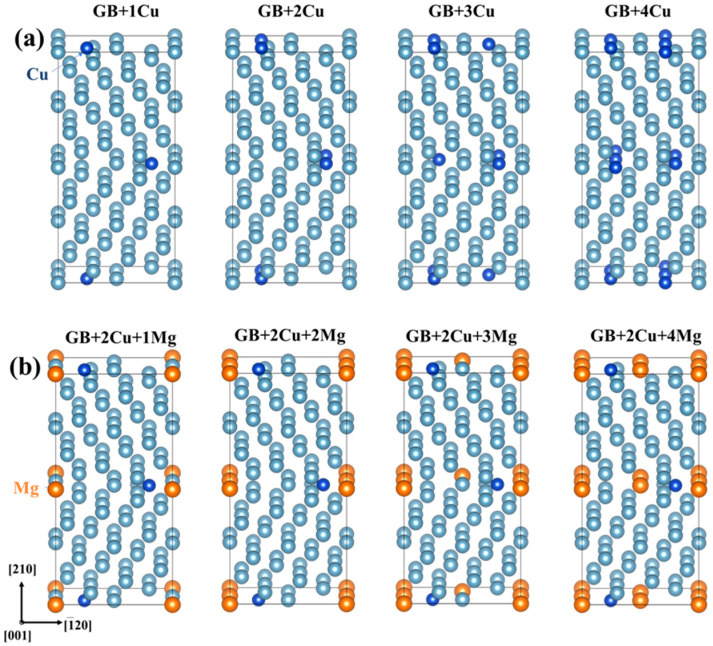
(**a**) The atomic diagram of different Cu-segregated Al Σ5(210) GBs, (**b**) an atomic diagram of different Mg-segregated GBs with 2 Cu segregations. The other Cu and Mg co-segregation GB models are constructed by the same method. The blue atom is Cu, and the orange atom is Mg.

**Figure 4 nanomaterials-14-01803-f004:**
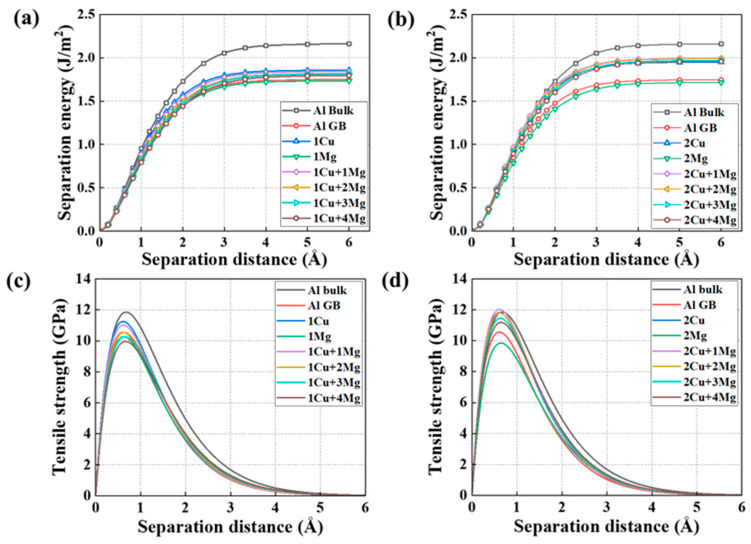
Separation energy of (**a**) 1 Cu + x Mg co-segregated Al Σ5(210) GBs and (**b**) 2 Cu + x Mg-segregated GBs at different separation distances. Tensile strength of (**c**) 1 Cu + x Mg co-segregated GBs and (**d**) 2 Cu + x Mg co-segregated GBs at different separation distances.

**Figure 5 nanomaterials-14-01803-f005:**
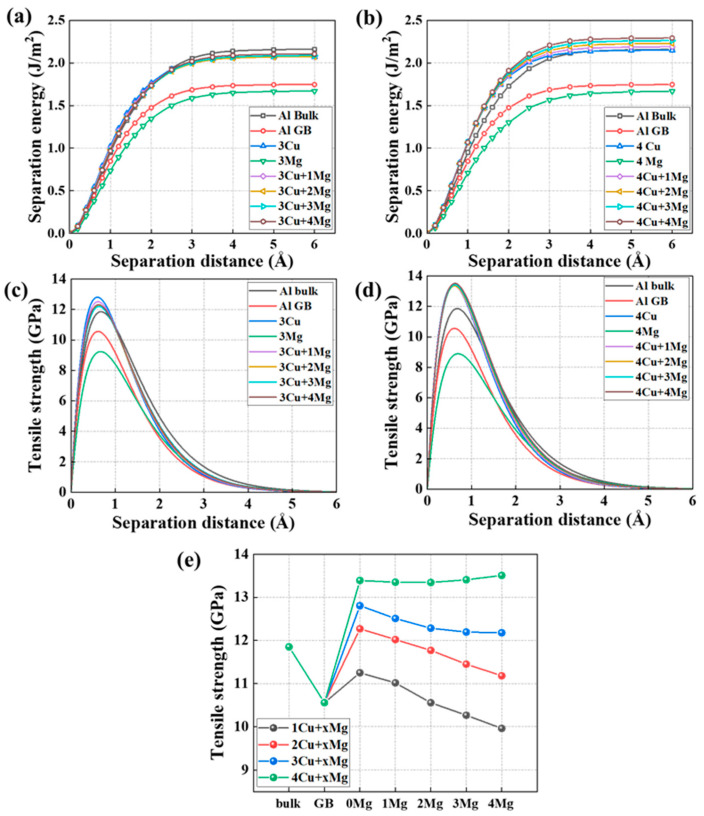
The separation energy of (**a**) 3 Cu + x Mg co-segregated Al Σ5(210) GBs and (**b**) 4 Cu + x Mg co-segregated GBs at different separation distances. The tensile strength of (**c**) 3 Cu + x Mg co-segregated GBs and (**d**) 4 Cu + x Mg co-segregated GBs at different separation distances. (**e**) The peak strength of the Al GB with different solutes’ co-segregation, the values of the Al (210) bulk model, and the pristine GB model are also plotted for comparison.

**Figure 6 nanomaterials-14-01803-f006:**
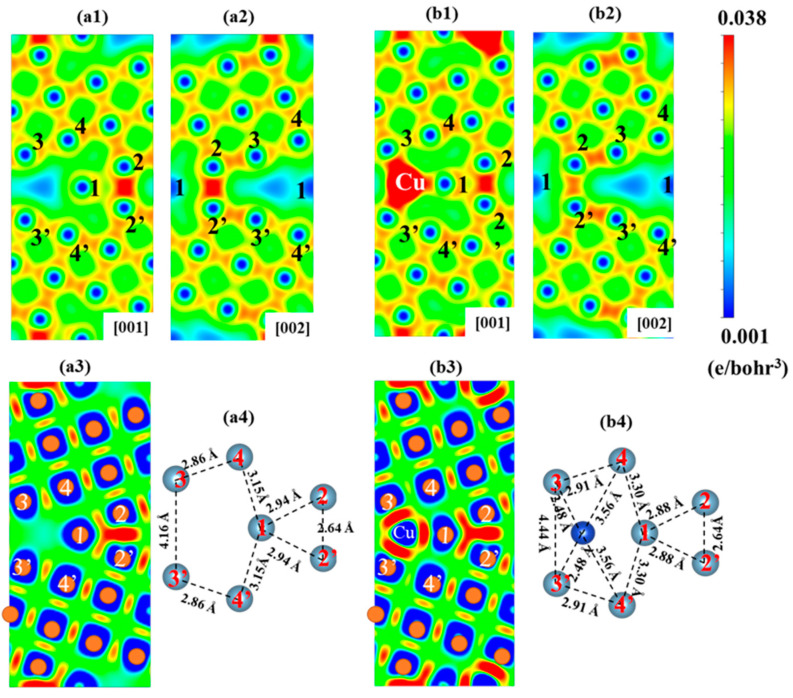
(**a1**,**a2**) The charge density distributions of the one Mg-segregated Al Σ5(210) GB in (001) and (002) planes, (**a3**,**a4**) the differential electron charge densities and GB structure of one Mg-segregated GB in the (001) plane. (**b1**,**b2**) The charge density distributions of the one Mg + two Cu-segregated Al GB in (001) and (002) planes, (**b3**,**b4**) the differential electron charge densities and GB structure of one Mg + two Cu-segregated GB in the (001) plane.

**Figure 7 nanomaterials-14-01803-f007:**
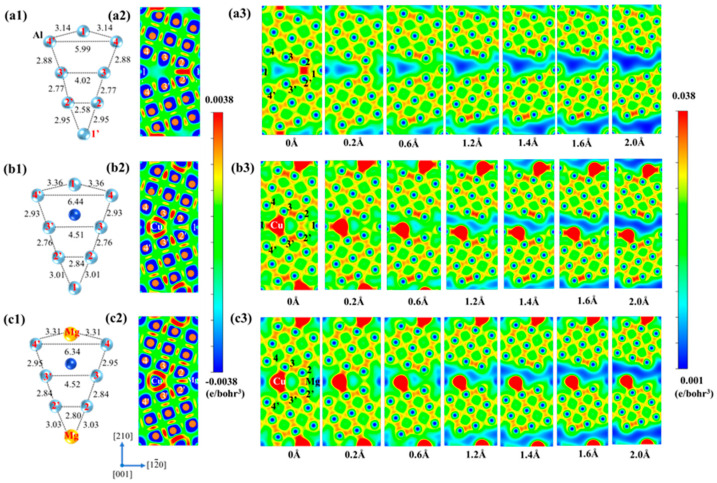
(**a1**–**c1**) The structure diagrams of the unsegregated Al Σ5(210) GB, 4 Cu-segregated Al GBs and 4 Mg + 4 Cu-segregated Al GB in (001) plane. (**a2**–**c2**) Differential electron charge densities distributions of the pristine Al GB, 4 Cu-segregated Al GBs, and 4 Mg + 4 Cu-segregated Al GBs in (001) plane. (**a3**–**c3**) The charity density distribution of the pristine Al GB, 4 Cu-segregated Al GBs, and 4 Mg + 4 Cu-segregated Al GBs in the (001) plane at different separation distances.

**Figure 8 nanomaterials-14-01803-f008:**
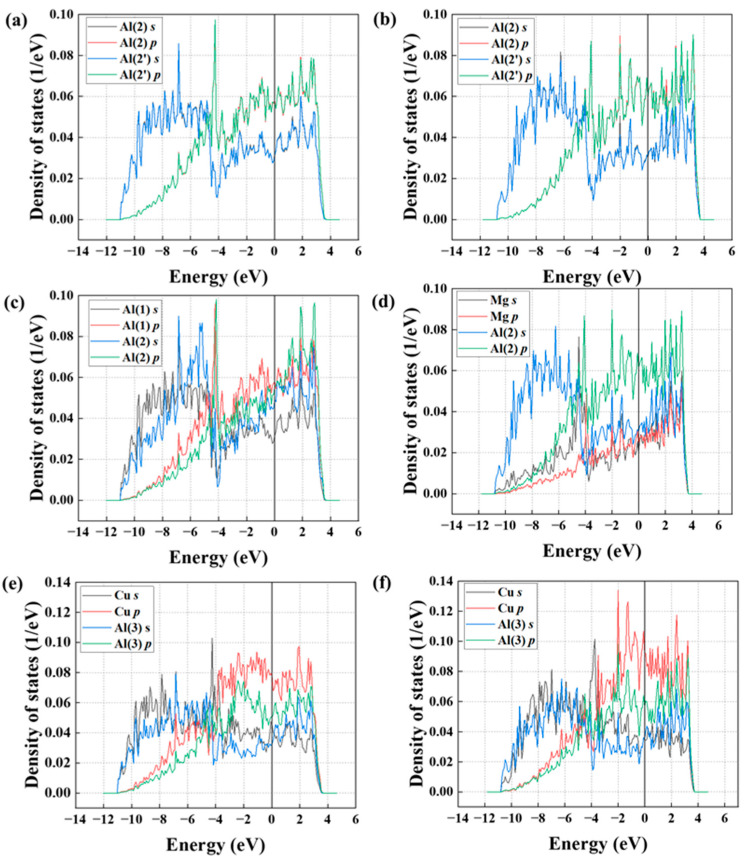
(**a**,**b**) The partial density of states (PDOS) for Al (2) and Al (2′) in both the four Cu-segregated GB and the four Cu + 4 Mg-segregated GB, (**c**,**d**) the PDOS for Al (1) and Al (2) in the four Cu-segregated Al GB and the PDOS for Mg and Al (2) at the four Cu + four Mg-segregated Al GBs, and (**e**,**f**) the PDOS for Cu and Al (3) for both four Cu-segregated Al GBs and four Cu + four Mg-segregated Al GBs. The Fermi level is represented by the line at 0 eV.

**Figure 9 nanomaterials-14-01803-f009:**
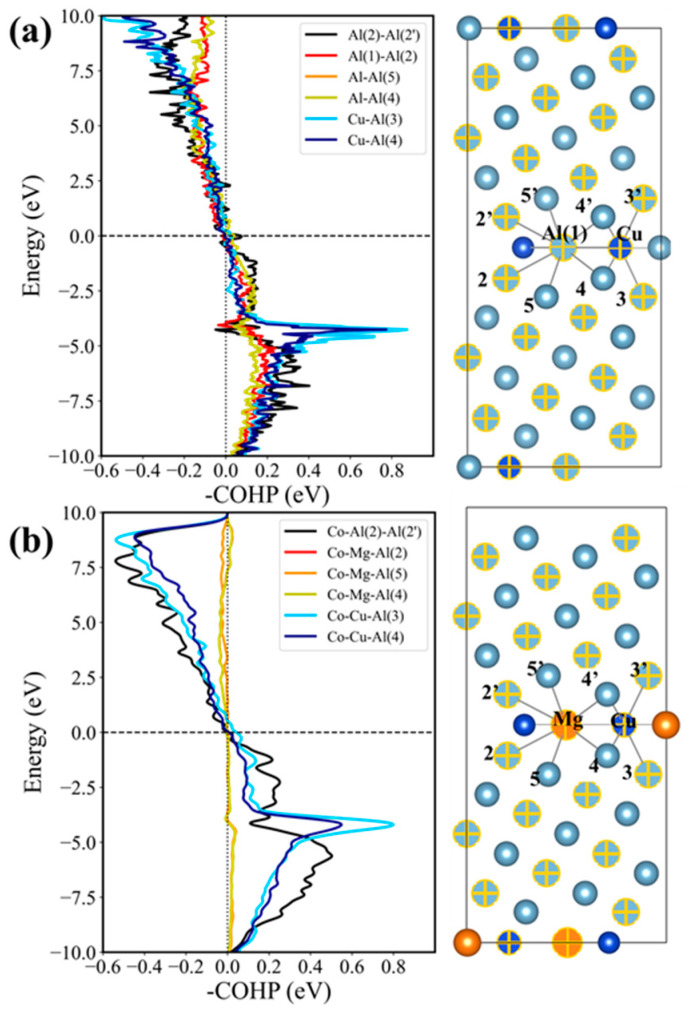
(**a**) Crystal Orbital Hamiltonian Populations (COHP) of the Cu-Al (3) bond, the Cu-Al (4) bond, etc., in the 4 Cu-segregated Al GBs. (**b**) The COHP of Mg-Al (2), Cu-Al (3), Cu-Al (4), etc., in the 4 Cu + 4 Mg co-segregated Al GB, where 0 eV in energy stands for the Fermi level.

**Table 1 nanomaterials-14-01803-t001:** The GB strength, fracture energy, and critical distance of different Mg and Cu co-segregated Al Σ5(210) GBs.

		0 Mg	1 Mg	2 Mg	3 Mg	4 Mg
0 Cu	Strength (GPa)	10.71	10.29	9.84	9.45	9.02
	Fracture energy (J/m^2^)	1.75	1.73	1.72	1.67	1.67
	Critical distance (Å)	0.612	0.619	0.643	0.667	0.691
1 Cu	Strength (GPa)	11.38	11.01	10.53	10.24	9.94
	Fracture energy (J/m^2^)	1.86	1.84	1.82	1.81	1.80
	Critical distance (Å)	0.612	0.613	0.637	0.649	0.667
2 Cu	Strength (GPa)	11.88	11.53	11.81	11.05	10.83
	Fracture energy (J/m^2^)	1.95	1.93	1.91	1.93	1.92
	Critical distance (Å)	0.612	0.619	0.619	0.637	0.649
3 Cu	Strength (GPa)	12.81	12.51	12.28	12.19	12.18
	Fracture energy (J/m^2^)	2.07	2.08	2.07	2.09	2.09
	Critical distance (Å)	0.594	0.613	0.619	0.625	0.631
4 Cu	Strength (GPa)	13.39	13.35	13.34	13.40	13.51
	Fracture energy (J/m^2^)	2.15	2.19	2.23	2.26	2.29
	Critical distance (Å)	0.589	0.601	0.613	0.619	0.625

**Table 2 nanomaterials-14-01803-t002:** The ICOHP values of the main bonds and the Bader charge values of the Al, Mg, and Cu atoms in both Al Σ5(210) GBs with 4 Cu and 4 Cu + 4 Mg segregation.

System	Bond	ICOHP (eV/Bond)	Atom	Bader Charge (e)	
				Atom in GB	Atom in Bulk
GB + 4 Cu	Al (1)-Al (2′)	−1.05	Al (1)	2.35	3.0
	Al (2)-Al (2′)	−1.65	Al (2)	2.55	3.0
	Cu-Al (3)	−1.75	Al (3)	2.73	3.0
	Al (1)-Al (4)	−1.01	Al (4)	2.55	3.0
	Al (1)-Al (5)	−0.95	Al (5)	2.73	3.0
	Cu-Al (4)	−1.59	Cu	13.10	11.0
GB + 4 (Cu + Mg)	Al (2)-Al (2′)	−2.42	Mg	0.52	2.0
	Mg-Al (2′)	−0.14	Al (2)	2.72	3.0
	Cu-Al (3)	−2.08	Al (3)	2.83	3.0
	Mg-Al (4)	−0.15	Al (4)	2.72	3.0
	Mg-Al (5)	−0.14	Al (5)	2.83	3.0
	Cu-Al (4)	−1.81	Cu	13.31	11.0

## Data Availability

The original contributions presented in this study are included in the article. Further inquiries can be directed to the corresponding authors.

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
