# Peer review of "The Effect of Solute Elements Co-Segregation on Grain Boundary Energy and the Mechanical Properties of Aluminum by First-Principles Calculation"

_nanomaterials, 2024, doi:10.3390/nano14221803_

Round 1
Reviewer 1 Report
Comments and Suggestions for Authors
1. As an important part of this paper, the term "solute elements" requires relevant references for clarification.
2. The term "grain boundary energy" requires incorporating findings from the abstract, results analysis, and conclusions.
3. The term "mechanical properties" is important to include in the conclusions section.
4. Given the significance of the "First-principles calculations" tool used by the authors, it is essential to include a key highlight in the conclusions.
5. In analyzing "GB energy," it is crucial for the authors to include the relevant findings in their conclusions.
6. Authors should include relevant references related to "GB strength" to enhance readability for the audience.
7. Equations 1, 2, 3, 4, and 5 are original creations of the authors; if not, please add the appropriate reference where it was published.
8. It is important to briefly comment on the results shown in Table 2 of ICOHP in the abstract.
9. It is essential to discuss the gaps in the conclusions regarding the physical term "PDOS."
10. Conclusions must be drawn from the findings on Al Σ5(210) GB in Figures 5, 6 and 7.
11. The data in Table 1 are part of this research or from published references; please cite them.
12. Minor changes suggested by the authors should prompt an English revision to preserve the paper's content.
Comments on the Quality of English LanguageMinor changes suggested by the authors should prompt an English revision to preserve the paper's content.
Author Response
We are grateful of the reviewers for their good suggestions to help us improve the work. We have revised the manuscript according to the reviewers' comments. Please find the point-to-point reply in the attached file.

Reviewer 2 Report
Comments and Suggestions for Authors
In this manuscript, the authors have studied an ab initio study of the effect of Mg and Cu co-segregation on the properties of Al Σ5(210) grain boundary (GB). The results showed that Mg has a preference for substitutional segregation that is far from Cu atoms. The results also showed that show that Mg and Cu co-segregation significantly decreases GB energy and enhances the stability of the Al Σ5(210) GB. I believe that this is an overall interesting study with findings that may be useful for the material design community. In my opinion, the manuscript could be accepted for publication after some minor improvements and clarifications:
1. Why does Mg tend to substitute the Al atom, as mentioned in lines 550 and 554? At least some explanation or discussion should be provided in that regard.
2. Authors stated that "According to the analysis of the binding energy between Mg and Cu, the primary factor driving Cu to facilitate Mg segregation is the energy of mutual attraction between them.". What is the nature of this interaction? That issue needs an additional explanation.
3. The paper is one of the many similar papers from the same research group and seems like fragmented work. The authors should clearly point out the differences between this study and previous studies from the same group.
Author Response

(The authors gave the same response as above.)
